# Extraction of Oil from *Allium iranicum* Seed and Evaluation of Its Composition and Quality Characteristics

**DOI:** 10.3390/foods14091483

**Published:** 2025-04-24

**Authors:** Abdolah Dadazadeh, Sodeif Azadmard-Damirchi, Zahra Piravi-Vanak, Mohammadali Torbati, Fleming Martinez

**Affiliations:** 1Department of Food Science and Technology, Faculty of Agriculture, University of Tabriz, Tabriz 51666, Iran; a.dadazadeh@tabrizu.ac.ir; 2Food, Halal, and Agricultural Products Research Group, Food Technology and Agricultural Products, Research Center, Standard Research Institute (SRI), Karaj 31745, Iran; 3Department of Food Science and Technology, Faculty of Nutrition, Tabriz University of Medical Sciences, Tabriz 15731, Iran; 4Pharmaceutical-Physicochemical Research Group, Department of Pharmacy, Faculty of Science, The National University of Colombia, Bogotá 11001, Colombia; fmartinezr@unal.edu.co

**Keywords:** composition, nutritional indices, oil extraction, oilseed, Iranian medicine, bioactive

## Abstract

The Allium plant genus has many species, among which *Allium iranicum* (AI) from the family Amaryllidaceae is endemic to Iran. There is no report on the oil composition of AI seeds. In this study, oil from AI seeds was extracted by a solvent and its composition and quality characteristics were determined. The yield of seed oil was 14.3%. The most predominant unsaturated fatty acid was linoleic acid (64.4%), followed by oleic acid (16.9%), and the main saturated fatty acids were palmitic acid (13.6%) followed by stearic acid (2.8%). Beta-sitosterol (50.7%), campestrol (15.7%), and delta5-avenasterol (8.2%) were the most dominant phytosterols in extracted AI oil. The most dominant tocopherol was α-tocopherol (1188 ppm) along with low amounts of δ- and γ-tocopherols. The obtained results showed that the oil extracted from seeds of AI can be a valuable by-product of this plant with suitable nutritional indices and can be used as a new source of vegetable oil. Further research is required to reveal its potential pharmaceutical and food applications.

## 1. Introduction

The Allium plant genus has about 900 species, among which *Allium iranicum* (AI) from the family Amaryllidaceae is a perennial plant endemic to Iran [1,2]. The AI plant produces small (3 mm in diameter), dark seeds (Figure 1).

Not many studies have been conducted on the AI plant, but it is used in traditional medicine for the treatment of headache, hemoptysis, asthma, bloating, hemorrhoids and obesity, and also as a tonic, diuretic, and stimulant [2,3]. This plant is also used in the treatment of vitiligo and chronic diarrhea [3]. The essential oil in the chives and stems of AI plants contains sulfur compounds and also contains alpha- and beta-pinene, which are effective against infectious endocarditis caused by Gram-positive bacteria such as *Staphylococcus aureus* and its toxin. Also, beta-pinene is used in pharmaceuticals and d-limonene is used in the cosmetics and health industries [4]. Due to its significant antimicrobial properties, this plant is placed in the category of Phytobiotics that can be used in the treatment of infections [2]. This plant is used as an anti-inflammatory compound in traditional medicine and its extract is taken orally as an antispasmodic and is used for controlling diabetes mellitus as well [4,5]. The ethanolic extract of AI has shown significant medicinal properties in controlling diabetes mellitus. This plant can prevent the breakdown of oligosaccharides and disaccharides by inhibiting the alpha-amylase enzyme, which reduces their absorption and prevents an increase in blood sugar [5]. AI oil is used to make hemorrhoid and diabetes treatments in traditional medicine. Also, a recent clinical study conducted on 80 people showed that the extract of this plant in the form of an ointment significantly reduced the severity of bleeding and itching, anal pain, and pain caused by defecation in these patients [6].

Different parts of this plant are used as a vegetable and the dried plant is popular as a spice and is used in the formulation of different food products as well [1]. The seeds of this plant have been studied for their essential oil compounds. However, its oil compounds have not been studied in terms of quantity and quality. Therefore, it is important to extract its oil as a new sources of vegetable oil, which can be helpful as an alternative to the conventional oilseeds. Also, new vegetable oils can give new opportunities to the pharmaceutical, cosmetics and food industries to use their bioactive components and positive effects to introduce new products to the market and consumers. Therefore, the aim of this study was to extract oil from AI seeds and evaluate its qualitative properties and compare it with similar and conventional vegetable oils.

## 2. Materials and Methods

### 2.1. Materials

AI seeds were purchased from a medicinal plant market (Tabriz, Iran). The chemicals used in this study were *n*-hexane, acetic acid, chloroform, sodium hydroxide, phenolphthalein, potassium hydroxide, sodium thiosulfate, and cyclohexane, which were all obtained from Merck & Co., Inc. (Darmstadt, Germany). Also, starch and Folin–Ciocalteu reagent were from Sigma-Aldrich Co. (St. Louis, MO, USA).

### 2.2. Methods

#### 2.2.1. Moisture Content

Seed moisture content was measured according to the method described by the American Chemical Society and calculated according to the following equation [7]:$$\mathrm{Moisture}\left(\%\right)=\frac{\left(W1-W2\right)}{W1}\times 100$$

W1 = weight of seeds before drying (in grams)

W2 = weight of seeds after drying (in grams)

#### 2.2.2. Oil Extraction from Seeds with Solvent

AI seeds were powdered with a mill (WJX-A750 model, China). Then, the seed powder was poured into a glass flask containing *n*-hexane solvent at a ratio of 1:10. For better mixing and penetration of the solvent, continuous shaking and mixing were performed in a shaker at an ambient temperature of 25 °C for 4 h. Then, the mixture was filtered and the solvent was evaporated by a rotary evaporator under vacuum at a temperature under 40 °C [8]. The remaining oil was stored in a freezer at −18 °C until further analysis.

#### 2.2.3. Oil Extraction Yield

The oil yield of AI seeds was calculated through the following equation [9].$$\mathrm{Oil}\;\mathrm{Yield}\;\left(\%\right)=\frac{M1}{M2}\times 100$$
where M1 is the mass of oil extracted from the seeds (in grams) and M2 is the mass of seeds (in grams).

#### 2.2.4. Specific Gravity

The specific gravity of the extracted oil at 25 °C was determined according to the AOCS method (Cc 25-7) [10].

#### 2.2.5. Refractive Index

The refractive index of oil samples was determined at 25 °C according to previously published methods [10]. 

#### 2.2.6. Acid Value

The acid value (AV) was determined by titration using the method described by the AOAC (Cd 3d-63) [11].

#### 2.2.7. Peroxide Value

The peroxide value (PV) was determined according to the official AOCS method (Cd 8-53) [10].

#### 2.2.8. Carotenoid Content

The extracted oil (7.5 g) was dissolved in cyclohexane (25 mL) and its absorbance was determined at wavelength of 450 nm using a spectrophotometer (UNICO model UV/Vis 2100) [12]. Then, the carotenoid content was calculated through the following equation:$$\mathrm{Cartenoid}\;\left(\mathrm{mg}/\mathrm{kg}\right)=\frac{\left(A\times 10^{6}\right)}{\left(2000\times 100\times d\right)}$$
where A is the absorption at 470 nm and *d* is the thickness of the spectrophotometer tube (which was 1 cm).

#### 2.2.9. Chlorophyll

For determination of the amount of chlorophyll, the extracted oil (7.5 g) was dissolved in cyclohexane (25 mL) and its absorbance was measured at 670 nm using a spectrophotometer (UNICO model UV/Vis 2100) [13]. Then, the chlorophyll content was calculated through the following equation:$$\mathrm{Chlorphyll}\;\left(\mathrm{mg}/\mathrm{kg}\right)=\frac{\left(A\times 10^{6}\right)}{\left(613\times 100\times d\right)}$$
where A is the absorption at 670 nm and *d* is the thickness of the spectrophotometer tube (which was 1 cm).

#### 2.2.10. Fatty Acid Profile

Fatty acid methyl esters (FAMEs) of the extracted oil were prepared via transesterification for fatty acid compositional determination [14]. Methanolic potassium hydroxide solution was added to the oil sample and kept at 60 °C for 10 min and then BF3 solution was added and kept at 60 °C for 10 min. The produced FAMEs were separated using *n*-hexane after centrifugation. The FAMEs were analyzed by gas chromatography (GC, Agilent Technologies, Santa Clara, CA, USA), coupled with a capillary column BPX70 (length of 50 m, inner diameter of 0.25 mm and particle size of the stationary phase of 25 μm). Helium was used as the carrier gas at a flow rate of 1 mL/min and a split/splitless injection system was used at a temperature of 230 °C. The temperature programming of the oven was as follows: first, the temperature was set at 158 °C for 5 min and then the temperature was raised to 230 °C at a rate of 2 °C/min [15]. A flame ionization detector was used at a temperature of 250 °C to detect FAMEs. The retention time was used in accordance with fatty acid standards for FAMEs identification. The area under the peaks was used for quantitative determination of the fatty acid percentages.

#### 2.2.11. Nutritional Indices

The following formulas were used to evaluate the extracted oil for atherogenicity index, thrombogenicity index, and hypocholesterolemic to hypercholesterolemic index (Hypo/HyperIndex) [16,17].$$\mathrm{Atherogenicity}\mathrm{Index}=\frac{C12:0+4\times C14:0+C16:0}{\Sigma \mathrm{MUFA}+\Sigma (\omega 6)+\Sigma (\omega 3)}$$$$\mathrm{Thrombogenicity}\mathrm{Index}=\frac{C14:0+C16:0+C18:0}{0.5\;x\;\Sigma \mathrm{MUFA}+0.5\;x\;\Sigma (\omega 6)+\Sigma (\omega 3)}$$$$\mathrm{Hypo}/\mathrm{HyperIndex}=\frac{C18:1+C18:2+C18:3+C18:4+C20:4}{C14:0+C16:0}$$

#### 2.2.12. Total Phenolic Compounds

Total phenolic compounds were measured using Folin–Ciocalteu reagent according to the method described by Caponio et al. (2015) [18]. For this purpose, absorption was determined at 765 nm using a spectrophotometer (UNICO UV/Vis 2100, Cambridgeshire, UK).

#### 2.2.13. Phytosterols

The phytosterol compounds in the oil were analyzed using gas chromatography (Agilent Technologies). For this purpose, the oil sample was saponified using 2 M ethanolic potassium hydroxide and the unsaponifiable fraction containing phytosterols was separated. Then, it was derivatized using trimethylsilyl reagent (TMS) and analyzed by GC. A fused-silica column (30 m long and 0.18 mm diameter, and the size of the stationary phase particles was 0.18 μm) was used for sterol separation, and helium and nitrogen were the carrier and make-up gases, respectively. The injector and flame ionization detector temperatures were 260 °C and 310 °C, respectively. The oven temperature was started at 60 °C, and then the temperature increased to 310 °C at a rate of 40 °C per min and was held for 27 min at 310 °C. The retention time was used to identify the phytosterols and 5α-cholestane was used as an internal standard to calculate the amount of phytosterols [19].

#### 2.2.14. Measurement of Tocopherols

The tocopherols composition and content were determined by HPLC using the method described by Azadmard-Damirchi and Dutta (2005) [16]. For this purpose, the oil sample (10 mg) was dissolved in *n*-heptane (1 mL). Then, 10 µL of the obtained mixture was injected into the HPLC. The column used for separation of the tocopherols was a LiChroCART (250-4 packed with LiChrosphere 100 NH_2_) (Merck KGaA, Darmstadt, Germany). Tocopherols were detected by a fluorescence detector (Walnut Creek, CA, USA) at a wavelength of 294 nm for excitation and 320 nm for emission. A mixture of *n*-heptane: tert-butyl methyl ether:tetrahydro-furan:methanol (79:20:0.98:0.02; *v*:*v*:*v*:*v*) was used as the mobile phase at a flow rate of 1.0 mL/min. Pure tocopherol isomers were used to identify the tocopherols and also for quantification via the external standard method.

#### 2.2.15. Triacyclglycerols

Triacylglycerols analysis was performed according to the method described by Farmani et al. (2006) [20] using GC (Agilent 7890 B), and peaks were identified based on retention time in comparison to standard triacylglycerols.

### 2.3. Statistical Analysis

Each experiment was repeated three times and the obtained data are presented as means ± standard deviations (SDs), which were calculated via Excel (Microsoft Excel 2019).

## 3. Results and Discussion

### 3.1. Physical and Chemical Properties

#### 3.1.1. Oil Percentage

One of the important factors in the evaluation of oilseed economical value is its oil content. According to the results obtained, AI seeds are 14.3% oil (Table 1), which is similar to grape seeds and pomegranate seeds. Oilseeds vary in their oil content from grapeseed (5.40–10.79%) [21] and pomegranate seed (10.8–15.0%) [22] to sunflower (40–45%) [21] and canola (35–40%) [23].

#### 3.1.2. Specific Gravity

Vegetable oils have characteristic specific gravities, which is a useful tool in their identification. This factor is commonly used in conjunction with other parameters during the evaluation and monitoring of the purity of vegetable oils [24]. According to the obtained result, the specific gravity of AI seed oil was 0.882 (Table 1), which is lower than the specific gravity of common vegetable oils such as sunflower oil (0.916–0.923), corn oil (0.917–0.925), and grapeseed oil (0.920–0.926) [25].

#### 3.1.3. Refractive Index

Each oil has a specific refractive index determined at a certain temperature, which is usually 20 °C for vegetable oils. Refractive index can be used to differentiate vegetable oils as well as being a suitable tool to evaluate vegetable oil quality and also shows the effect of different processes on the quality of vegetable oils [26]. The refractive index of AI seed oil was 1.468, which is similar to safflower (1.467–1.470), corn (1.465–1.468), sunflower (1.461–1.475), and grapeseed (1.467–1.477) oils [25].

#### 3.1.4. Acid Value

Acid value shows the level of free fatty acid content in vegetable oils. Free fatty acids are formed by triacylglycerol hydrolysis, which occurs via enzyme activity or thermal stress. A low content of free fatty acid is favorable as it makes oils more stable during storage and during food processing [27]. Acid value is one of the main quality factors that is examined in edible oils to determine its censurability. The acid value of the AI seed oil was 1.6 mg KOH/g oil, which is an acceptable level. The acid value of AI seed oil is similar to sesame oil (1.34 mg KOH/g oil) and lower than palm oil (2.64 mg KOH/g oil) but higher than mustard (0.56 mg KOH/g oil), soybean (0.35 mg KOH/g oil), and coconut (0.48 mg KOH/g oil) oils [28]. The acid value of vegetable oils should be lower that 4 mg KOH/g oil according to the Codex Alimentarius standard, which shows that the extracted oil has high quality in terms of its acid value.

#### 3.1.5. Peroxide Value

Peroxides are primary oxidation products that are formed in the initial stage of vegetable oil oxidation and are measured by PV. A lower PV means a higher quality of vegetable oil, as peroxides have adverse effects on oil quality and limits their applications and uses in food product preparation and formulations [29]. The PV of the AI seed oil was 2 (meqO_2_/kg oil), which is low for a solvent extracted vegetable oil [30]. Comparing the peroxide of AI seed oil with some conventional oils such as mustard (3.08 meqO_2_/kg oil) [31], soybean (4.55 meqO_2_/kg oil) [32] and sunflower (3.12 meqO_2_/kg oil) [33] shows that it has a lower peroxide content and this can show its suitable oxidative stability. However, crude oils with a higher level of PV can be refined to reduce their PV and also their AV. A low level of PV for an extracted oil can be due to the high quality of the raw material (seeds), good storage conditions, low levels of oxidative enzymes such as lipoxygenases, and high stability of the oil, which is influenced by its fatty acid composition and its antioxidative component contents [34].

#### 3.1.6. Carotenoid Content

Carotenoids are minor components present in vegetable oils and have different roles. Carotenoids are bioactive components and products of vitamin A and also have antioxidative properties, which enhance vegetable oils’ oxidative stability [35]. Their suitable content in vegetable oils is important from nutritional and technological points of view [36]. The carotenoid content of AI seed oil was 3.2 ppm, which is much lower than oils such as rapeseed (63.60) [31], grapeseed (67.99) [37], soybean (24.96) [32] and sunflower (21.20) [33].

#### 3.1.7. Chlorophyll Content

Chlorophylls give a green color to vegetable oils and can also affect their oxidative stability. They can reduce the oxidative stability of vegetable oils during exposure to light via photooxidation; however, this does not affect the oxidative stability of vegetable oils in dark conditions. In some cases, there is a need for refining and especially bleaching to remove the chlorophyll from vegetable oils [38]. AI seed oil has 6.2 ppm chlorophyll, which is comparable to vegetable oils such as sunflower (5.62) and soybean (2.43) [32,33]. However, it has less chlorophyll than grapeseed oil (17.04) and rapeseed oil (26.00) [37].

#### 3.1.8. Total Phenol Content

Phenol content and composition can affect the nutritional properties and stability of a vegetable oil against oxidation. Vegetable oils rich in phenolic components such as virgin olive oils are preferred in the diet due to their positive health effects, and they are a component of the Mediterranean diet [39]. The total phenol content of AI seed oil was 6.2 (mg/kg oil), which was higher than mustard oil (5.6 mg/kg oil), sunflower oil (4.9 mg/kg oil) and sesame oil 3.3 (mg/kg oil) and lower than of groundnut oil (30.9 mg/kg oil), coconut oil (18 mg/kg oil) and rice bran oil (8.9 mg/kg oil) [40]. However, virgin olive oil is one of the vegetable oils that has the highest total phenol content. It was reported that the total phenol content of different Italian extra-virgin olive oils can vary from 97.3 to 573.2 (mg/kg oil) depending on the geographical areas the fruits are harvested from [41].

#### 3.1.9. Fatty Acids

Fatty acids have different roles in vegetable oils, such as stability, nutritional value and also technological properties [42,43]. The predominant fatty acids in AI oil were linoleic acid (64.4%), followed by oleic acid (16.9%), palmitic acid (13.6%), stearic acid (2.8%) and other fatty acids at minor levels (Table 2). According to the obtained results, AI seed oil is classified in the linoleic acid group. High level of polyunsaturated fatty acids can affect and reduce oil oxidative stability [44]. According to the obtained results, this oil has a fatty acid composition similar to safflower, corn, sunflower and grapeseed oils [24].

According to this research, it was found that linoleic acid is the most dominant fatty acid in AI seeds. This fatty acid is one of the essential fatty acids that the human body is unable to synthesize. Its deficiency causes skin lesions and scaling, changes in fatty acids in plasma, growth delay and thrombocytopenia [45]. However, excessive consumption of linoleic acid can also cause health concerns. According to researchers, the ideal ratio of linoleic acid (as an ω_6_ fatty acid) to linolenic acid (as an ω_3_ fatty acid) is 1:1 and a maximum of 4:1 [44,46]. Linoleic acid is a precursor of arachidonic acid [47] and arachidonic acid is also a precursor of inflammatory compounds such as prostaglandin and leukotrienes [47,48], which can promote cardiovascular diseases such as atherosclerosis, cancers and inflammatory diseases [49]. Therefore, there is a need to blend this oil with other vegetable oils rich in ω_3_ or ω_9_ fatty acids in a suitable and balanced diet and also to obtain an oil with proper oxidative stability as well.

### 3.2. Nutritional Quality Index

The nutritional quality indices of AI seed oil and other similar oils in terms of fatty acid profile were calculated and are presented in Table 3.

#### 3.2.1. Atherogenicity Index

The atherogenicity index of oils indicates that an oil can cause atherosclerosis and can increase or decrease the risk of heart attack [50]. This index of oils actually shows the ratio of saturated fatty acids to unsaturated fatty acids. The lower the value of this index, the healthier the oil and the lower the risk of atherosclerosis [50]. The atherogenicity index of *A. iranicum* oil is similar to corn and sunflower oil and much lower than that of dairy fat (1.37–5.02) [51]. This result shows that this oil is suitable to be in the daily diet from an atherogenicity index point of view, as it has higher content of unsaturated fatty acids compared to its saturated fatty acid content.

#### 3.2.2. Thrombogenicity Index

Research has shown a close relationship between the consumption of certain saturated fatty acids and platelet aggregation, followed by clot formation and heart and stroke. Animal studies have shown that following the injection of stearic and myristic fatty acids, significant clots form in the arteries of dogs [52]. The higher the amount of myristic, stearic, and palmitic saturated fatty acids in the oil, the greater the likelihood and extent of clot formation. The thrombogenic index actually examines and evaluates the potential of oils to form clots [53]. This index shows the ratio of saturated fatty acids, mainly myristic acid, palmitic acid and stearic acid, to unsaturated fatty acids, oleic acid, linoleic acid and alpha-linolenic acid. The lower the index, the healthier the oil and the lower the probability of thrombosis [54]. The thrombogenic index of *A. iranicum* oil was 0.23, which is similar to the thrombogenic index of corn oil and higher than safflower oil and grape seed oil, but lower than sunflower oil. Of course, the thrombogenicity index of *A. iranicum* oil is lower than that of dairy fat (up to 5.04) [55,56] and meat fat (0.28–1.69) [50].

#### 3.2.3. Hypocholesterolemic to Hypercholesterolemic Index (Hypo/HyperIndex)

The hypocholesterolemic to hypercholesterolemic index of an oil predicts the effect of that oil on the amount and type of cholesterol in the blood. This index shows the contents of unsaturated fatty acids such as oleic acid, linoleic and linolenic acid to that of saturated fatty acids such as myristic acid and palmitic acid [57]. The higher the Hypo/HyperIndex, the better cholesterol (HDL) the oil produces in the body, which has a protective effect on the heart and blood vessels, and the lower the Hypo/HyperIndex, the more bad cholesterol (LDL) is produced in the body, which increases the risk of cardiovascular diseases [58]. The Hypo/HyperIndex of *A. iranicum* oil compared to similar oils in terms of fatty acid profile shows that this oil is better than sunflower oil and similar to corn oil, but it is lower than grapeseed and safflower oil [24].

### 3.3. Phytosterols

Phytosterols are a fingerprint of vegetable oils as they differ in composition and content and therefore can be used to detect vegetable oil authenticity and adulteration [59]. They are also important in the oxidative stability and have different positive health effects [60]. For example, delta5-avenasterol has antioxidative properties and lupeol has antitumor effects; also, all phytosterols in the proper amount in the daily diet can decrease cholesterol levels and reduce the risk of cardiovascular diseases [61].

It has been reported that phytosterols can compete with cholesterol in the process of digestion and absorption and reduce cholesterol absorption. Research shows that consumption of 2–3 g of phytosterols daily can reduce the absorption of bad cholesterol (LDL type) by about 10% in the bloodstream and reduce the risk of cardiovascular diseases [62]. There is also a competition between types of sterols; for example, campestrol is better absorbed than sitosterol [63]. In general, phytosterols have very useful health and nutritional properties and are considered a type of nutraceutical. Beta-sitosterol is the predominant phytosterol in AI seed oil, accounting for 50.72% among the other seven types of phytosterols. Studies have shown that this phytosterol reduces the inflammatory process of fatty liver caused by fat accumulation by reducing the amount of reactive oxygen species (ROS) [64]. On the other hand, phytosterols can show anti-cancer activity by inducing apoptosis and reducing metastasis, and by reducing the amount of ROS, they prevent the oxidative stress of cells [65] and stimulate the immune system against cancer cells [66].

Different types of phytosterols were identified in AI seed oil. Beta-sitosterol was the main phytosterol followed by campestrol, delta-5-avenasterol and stigmasterol along with some other sterols at lower amounts (Table 4).

The results showed that AI seed oil had a phytosterol content (2850 mg/kg oil) comparable with other vegetable oils such as safflower oil (2100–4600 mg/kg oil), flaxseed oil (2300–6900 mg/kg oil), grapeseed oil (2000–7000 mg/kg oil), and sweet almond oil (1590–4590 mg/kg oil). Also, the phytosterols of AI seed oil are higher than the phytosterols of coconut oil (400–1200 mg/kg oil), hazelnut oil (1200–1800 mg/kg oil), palm oil (300–700 mg/kg oil), and walnut oil (500–1760), but are lower than the phytosterol content of corn oil (7000–22,100 mg/kg oil), sesame oil (4500–19,000 mg/kg oil), and rice bran oil (10,500–31,000 mg/kg oil) [24].

### 3.4. Tocopherols

Tocols are divided into two types, tocopherols and tocotrienols, both of which have alpha, beta, gamma and delta isomers that are found in vegetable oils [67]. Tocols can have vitamin E activity along with being powerful antioxidants [68]. Generally, the main source of vitamin E in the diet is vegetable oil, such as olive oil, sunflower, soybean and almond oil [69]. AI seed oil is also considered a good source of vitamin E. Three types of tocopherols, α-, γ-, and δ-tocopherols, were identified in AI seed oil, in which α tocopherol was the main compound followed by γ-tocopherol and δ-tocopherol (Table 5). This research showed that AI seed oil is a rich source of α-tocopherol (1188 ppm), which is very high compared with many other vegetable oils [24].

Tocopherols play a very important role in inhibiting the oxidization of vegetable oils and prevent them from developing an unpleasant aroma and taste [70]. In general, the concentration of tocopherols in vegetable oils is between 200–1000 ppm [71]. Sunflower oil with 690 ppm and hazelnut with 573 ppm of α-tocopherol are considered rich sources of α-tocopherol, but this research shows that AI seed oil is a richer source of α-tocopherol compared to these two vegetable oils [24].

α-Tocopherol is of particular importance among the various tocopherol isomers [72]. α-Tocopherol has stronger antioxidant and antitumor effects than other tocopherols [73,74]. The most important feature of α-tocopherol is its anti-inflammatory properties, which can alleviate a wide range of inflammatory diseases such as cardiovascular diseases [75], cancers, and Alzheimer’s [76], and can prevent ischemic heart diseases by inhibiting LDL oxidation [77] and prevention of the adhesion of platelets [78].

## 4. Conclusions

This is the first report on the oil content and oil composition of AI seeds, which can be considered as a new source of vegetable oil with a suitable oil extraction yield (14.3%). This oil belongs to the linoleic–oleic group of vegetable oils. In terms of composition, AI seed oil is similar to oils such as safflower, sunflower, cottonseed and grape seed. This oil is a rich source of bioactive components such as phytosterols and tocopherols, particularly α-tocopherol. Therefore, oil can be a valuable by-product of *Allium iranicum*. It needs more investigation on its nutritional value, positive health properties and its application in the daily diet. AI seed oil, due to its richness in α-tocopherol compared to other oilseeds, can be a good source of this bioactive component through industrial cultivation. Also, there is a need for further research to evaluate this oil via toxicological and mutagenicity studies.

## Figures and Tables

**Figure 1 foods-14-01483-f001:**
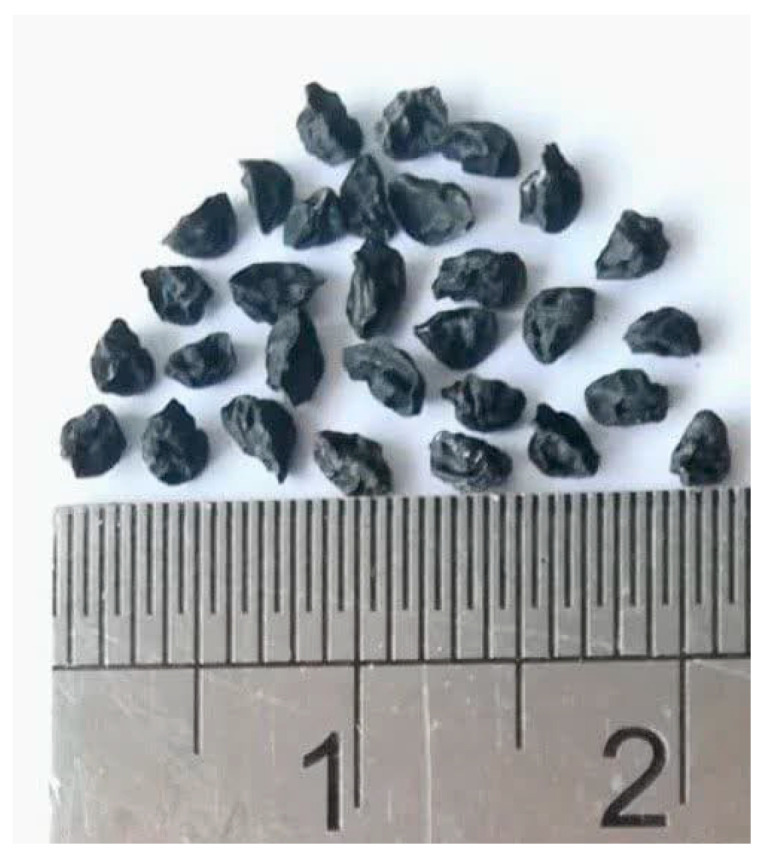
*Allium iranicum* seeds: appearance and size (in cm).

**Table 1 foods-14-01483-t001:** Physical and chemical characteristics of *Allium iranicum* seed oil.

Parameter	Results
Oil percent (%)	14.3 ± 0.2 *
Specific gravity	0.882 ± 0.0
Refractive index	1.468 ± 0.0
Acid value (mg KOH/g oil)	1.6 ± 0.1
Peroxide value (meq O_2_/kg oil)	2.0 ± 0.1
Carotenoid content (mg/kg oil)	3.2 ± 0.1
Chlorophyll content (mg/kg oil)	6.1 ± 0.3
Total phenolic compounds (mg/kg oil)	6.2 ± 0.5

* Mean is shown as ±SD.

**Table 2 foods-14-01483-t002:** Fatty acid (%) profile of *Allium iranicum* seed oil in comparison with other vegetable oils with a similar fatty acid composition.

Fatty Acid	*A. iranicum* Seed Oil	Safflower Oil	Corn Oil	Sunflower Oil	Grapeseed Oil
Lauric acid	0.2 ± 0.01 *	ND **	ND	ND	ND
Myristic acid	0.6 ± 0.10	ND-0.2	ND	ND-0.2	ND-0.3
Palmitic acid	13.6 ± 0.71	5.3–8.0	8.6–16.5	21.4–26.4	5.5.11.0
Palmitoleic acid	0.3 ± 0.01	ND-1.2	ND-0.5	ND-0.3	ND-1.2
Stearic acid	2.8 ± 0.23	1.9–2.9	ND-3.3	2.7–6.5	3.0–6.5
Oleic acid	16.9 ± 1.03	8.4–21.3	20.0–42.2	14.0–43.0	12.0–28.0
Linoleic acid	64.4 ± 2.80	67.8–83.2	34.0–65.6	45.4–74.0	58.0–78.0
α-linolenic acid	0.7 ± 0.10	ND-0.1	ND-2.0	ND-0.3	ND-1.0
Arachidic acid	0.3 ± 0.01	0.2–0.4	0.3–1.0	0.1–0.5	ND-1.0
Behenic acid	0.1 ± 0.01	ND-1.0	ND-0.5	0.3–1.5	ND-0.5

* Mean is shown as ±SD. ** ND = not detectable.

**Table 3 foods-14-01483-t003:** Nutritional quality indices (NQI) of *Allium iranicum* oil and its comparison with other similar oils in terms of fatty acid profile.

NQI	*A. iranicum*	Safflower	Corn	Sunflower	Grapeseed
Atherogenicity	0.20	0.08	0.15	0.28	0.10
Thrombogenicity	0.23	0.11	0.21	0.27	0.16
Hypo/Hyper *	5.77	13.33	6.53	3.68	10.53

* Hypocholesterolemic to Hypercholesterolemic.

**Table 4 foods-14-01483-t004:** Phytosterol composition (%) of *Allium iranicum* seed oil in comparison with some other vegetable oils.

Phytosterol	*A. iranicum*	Safflower [24]	Corn [24]	Sesame [24]
Brassica sterol	2.5 ± 0.1 *	ND **–0.3	ND-0.2	0.1–0.2
Campesterol	15.7 ± 1.2	5.0–13.0	16.0–24.1	10.1–20.0
Stigmasterol	4.3 ± 0.5	4.5–13.0	4.3–8.0	3.4–12.0
Beta-sitosterol	50.7 ± 3.7	42.0–70.0	54.8–66.6	57.7–61.9
Delta-5-avenasterol	8.2 ± 1.4	1.5–6.9	1.5–8.2	6.2–7.8
Delta-7-stigmastanole	1.3 ± 0.1	6.5–24.0	0.2–4.2	0.5–7.6
Delta-7-avenastero	0.8 ± 0.1	ND–9.0	0.3–2.7	1.2–5.6
Unknown or other	16.4 ± 0.5	3.5–9.5	ND-2.4	0.7–9.2
Total (mg/kg oil)	2850 ± 49	2100–4600	7000–22,100	4500–19,000

* Mean is shown as ±SD. ** ND = not detectable.

**Table 5 foods-14-01483-t005:** Tocopherol composition of *Allium iranicum* seed oil.

Tocopherol	Content (ppm)
α-tocopherol	1188.0 ± 11.5 *
γ-tocopherol	10.9 ± 2.0
δ-tocopherol	1.7 ± 0.3

* Mean is shown as ±SD.

## Data Availability

Detailed data are available upon request from the authors. The data are not publicly available due to privacy restrictions.

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
