# Peer review of "Extraction of Oil from Allium iranicum Seed and Evaluation of Its Composition and Quality Characteristics"

_foods, 2025, doi:10.3390/foods14091483_

Round 1
Reviewer 1 Report
Comments and Suggestions for Authors
This manuscript analyzes the composition and quality attributes of oil derived from seeds of Allium iranicum.
Some corrections should be introduced:
Section 2.2.2
Clarify the meaning of "t" in the sentence: "The oil was stored at freezer 1t -18°C until further analysis."
Section 2.2.10
The abbreviation "AI" is used for both Allium iranicum and atherogenicity index. These should be distinguished to avoid confusion.
Section 2.3.
„Each experiment was repeated three times and the obtained data were reported as mean.” Simply reporting the mean without any measure of variability (e.g., standard deviation or standard error) or statistical tests does not provide sufficient statistical insight. Statistical analysis should be completed.
Section 3.1.1
Include a reference to Table 1 in the text.
The text analyzes the refractive index, but the Methods section does not mention this parameter. It should be added.
To facilitate comparison, provide the refractive indexes of the oils mentioned (safflower, corn, sunflower, and grapeseed).
Similarly, for comparative purposes, include the acid values of other oils in the text.
The peroxide value should also be compared with values from other oils.
Editorial errors
Why is the section numbered 3.1.1 when there is no 3.1.2, only 3.2? The numbering should be corrected.
In section 3.1.1, two different font sizes are used. Ensure formatting consistency.
Section 3.2
Report the carotenoid content in various oils, including rapeseed, grapeseed, soybean, and sunflower.
Section 3.3
Report the chlorophyll content in sunflower, soybean, rapeseed, and grapeseed oils.
Section 3.6.
Include a reference to Table 3 in the text.
The atherogenicity index of dairy fat should be provided for comparison.
Section 3.6.2
The manuscript states: "Of course, the atherogenic index of A. iranicum oil is lower than that of dairy fat (0.39-5.04) [54, 55] and meat fat (0.28-1.69) [50]." However, this section discusses the thrombogenicity index. Clarify whether the values refer to the atherogenicity index or the thrombogenicity index.
Section 3.6.3
The abbreviation for the hypocholesterolemia to hypercholesterolemia index is inconsistently used as H/H in this section and HI in chapter 2.2.10. This should be corrected for consistency.
Section 3.6 (Phytosterols)
The Phytosterols subsection should have a separate numbering, as 3.6 is designated for the Nutritional Quality Index.
Table 4
A bibliography should be provided for the data presented in Table 4.
Author Response
Dear Reviewer,
We would like to thank you very much for your valuable comments and suggestions which were very useful and helpful to improve the manuscript quality. We revised the manuscript extensively according to the comments and suggestions. Changes were made in red to be easily followed.
Comments and Suggestions for Authors
This manuscript analyzes the composition and quality attributes of oil derived from seeds of Allium iranicum.
Some corrections should be introduced:
Comment 1: Section 2.2.2
Clarify the meaning of "t" in the sentence: "The oil was stored at freezer 1t -18°C until further analysis."
Response 1: It should be " The oil was stored at freezer at -18°C until further analysis." It was corrected.
Comment 2: Section 2.2.10
The abbreviation "AI" is used for both Allium iranicum and atherogenicity index. These should be distinguished to avoid confusion.
Response 2: The "AI" abbreviation was used for Allium iranicum and it was deleted for atherogenicity index.
Comment 3: Section 2.3.
„Each experiment was repeated three times and the obtained data were reported as mean.” Simply reporting the mean without any measure of variability (e.g., standard deviation or standard error) or statistical tests does not provide sufficient statistical insight. Statistical analysis should be completed.
Response 3: It was revised in the text and in the Tables as advised.
Comment 4: Section 3.1.1
Comment 4.1: Include a reference to Table 1 in the text.
Response 4.1: References were included in Table 1 as advised.
Comment 4.2: The text analyzes the refractive index, but the Methods section does not mention this parameter. It should be added.
Response 4.2: Refractive Index was added to the methods as section 2.2.5.
Comment 4.3: To facilitate comparison, provide the refractive indexes of the oils mentioned (safflower, corn, sunflower, and grapeseed).
Similarly, for comparative purposes, include the acid values of other oils in the text.
The peroxide value should also be compared with values from other oils.
Response 4.3: The refractive index, peroxide value and acid value results were compared with other vegetable oils and included in the text as advised.
Comment 5: Editorial errors
Why is the section numbered 3.1.1 when there is no 3.1.2, only 3.2? The numbering should be corrected.
In section 3.1.1, two different font sizes are used. Ensure formatting consistency.
Response 5: Sorry for these errors in the formatting. All were corrected.
Comment 6: Section 3.2
Report the carotenoid content in various oils, including rapeseed, grapeseed, soybean, and sunflower.
Response 6: The carotenoid content in various oils were included as advised.
Comment 7: Section 3.3
Report the chlorophyll content in sunflower, soybean, rapeseed, and grapeseed oils.
Response 7: The chlorophyll content of various oils was included as advised.
Comment 8: Section 3.6.
Include a reference to Table 3 in the text.
The atherogenicity index of dairy fat should be provided for comparison.
Response 8: Table 3 was referenced in the text and the data on atherogenicity of dairy products were also written as recommended.
Comment 9: Section 3.6.2
The manuscript states: "Of course, the atherogenic index of A. iranicum oil is lower than that of dairy fat (0.39-5.04) [54, 55] and meat fat (0.28-1.69) [50]." However, this section discusses the thrombogenicity index. Clarify whether the values refer to the atherogenicity index or the thrombogenicity index.
Response 9: It was a typing error and according to the comment, the word atherogenicity was corrected to thrombogenicity
Comment 10: Section 3.6.3
The abbreviation for the hypocholesterolemia to hypercholesterolemia index is inconsistently used as H/H in this section and HI in chapter 2.2.10. This should be corrected for consistency.
Response 10: This index was also standardized throughout the article with the abbreviation Hypo/Hyper Index.
Comment 11: Section 3.6 (Phytosterols)
The Phytosterols subsection should have a separate numbering, as 3.6 is designated for the Nutritional Quality Index.
Response 11: It was done as recommended.
Table 4
A bibliography should be provided for the data presented in Table 4.
Response 11: References were provided for Table 4.

Reviewer 2 Report
Comments and Suggestions for Authors
Extraction of Oil from Allium iranicum Seed and Evaluation of Its Composition and Quality Characteristics.
Introduction
(Line 31-33) The botanical description of species should be used references from botany books in the area, not articles.
The introduction needs to be improved structure, as the information is mixed, for better organization in the text I suggest that you start this topic by reporting the botanical importance, traditional uses, chemical composition of its oils and end with the objective of the research. Because it is very poor in information.
Line 47. There are numerous edible and non-edible applications. What applications?? Explain better?
Results and Discussion
Line 165. Observe the formatting of the paragraph.
14.3% oil. Is this content significant? Explain?
Line 191-195 is unformatted.
The carotenoid content of AI seed oil is 3.2 ppm. Is this content significant from a nutritional point of view?
Line 330. According to previous studies. What are these studies? What do they report?
Conclusion
This topic is written as a presentation of results, and needs to be rewritten, what is the importance of this study? Prospects and future studies?
Line 343. Allium Iranicum. Check the formatting of the scientific name of botanical species. Beside the text there are errors, and it should be noted.
Author Response
Dear Reviewer,
We would like to thank you very much for your valuable comments and suggestions which were very useful and helpful to improve the manuscript quality. We revised the manuscript extensively according to the comments and suggestions. Changes were made in red to be easily followed.
Introduction
Comment 1: (Line 31-33) The botanical description of species should be used references from botany books in the area, not articles.
Response 1: Botany book reference wad included for description as advised.
Wendelbo, P. 1971. Alliaceae. In: Rechiger, K.H.(ed.), Flora Iranica, Tomus 76, Akademische Druck und Ver-lagsanatalt Graz
Comment 2: The introduction needs to be improved structure, as the information is mixed, for better organization in the text I suggest that you start this topic by reporting the botanical importance, traditional uses, chemical composition of its oils and end with the objective of the research. Because it is very poor in information.
Response 2: Thank you very much for this valauble comments. The introduction part was revised as advised.
Comment 3: Line 47. There are numerous edible and non-edible applications. What applications?? Explain better?
Response 3: It was included as advised.
Results and Discussion
Comment 1: Line 165. Observe the formatting of the paragraph.
Response 1: Formatting was done as advised.
Comment 2: 14.3% oil. Is this content significant? Explain?
Response 2: This amount is not significant compared to seeds that contain 40 to 50 percent oil, but, there is a need for further research to exploar its pharmacutical and other beneficial effects which can enhance its economical value as well..
Comment 3: Line 191-195 is unformatted.
Response 3: Formatting done.
Comment 4: The carotenoid content of AI seed oil is 3.2 ppm. Is this content significant from a nutritional point of view?
Response 4: Carotenoids are nutritionally and technologically important, but since the amount of carotenoids in Allium iranicum oil is low, its nutritional importance will also be low. However, given that the aim of this article was to identify the characteristics of Allium iranicum oil, an attempt has been made to identify and introduce the various components of the oil.
Comment 5: Line 330. According to previous studies. What are these studies? What do they report?
Response 5: Thank you for your careful reading of the text and your very thoughtful comment. Previous studies refer to the material that other researchers have published on alpha-tocopherol in other articles, but in order to easily convey the meaning of the text to the reader, the phrase "previous study" was removed from the paragraph.
Conclusion
Comment 1: This topic is written as a presentation of results, and needs to be rewritten, what is the importance of this study? Prospects and future studies?
Response 1: The importance of the study, perspectives, and future studies were added to the conclusion section as recommended.
Comment 2: Line 343. Allium Iranicum. Check the formatting of the scientific name of botanical species. Beside the text there are errors, and it should be noted.
Response 2: The text was reviewed and revised as recommended.

Reviewer 3 Report
Comments and Suggestions for Authors
The article is about the composition of Iranian garlic seeds. The study reveals the nutritional value and potential uses of garlic seeds. My comments:
Introduction:
- Is the extraction of Allium seed oil economically viable?
- It is also worth mentioning the common word "garlic".
Materials and methods:
- Was a program such as Excel or SPSS used?
Results and discussion:
- It's unclear on line 166, what does "(1)" and the moved text mean?
- The text font is different (lines 191-195).
- 3.2. section: Please provide the amount of other carotenoids in plants, not just a link to the source.
- Also 3.3. section, provide values of chlorophyll content not only link to source.
- Review all results and discussion and supplement with numerical values ​​from other sources, not just a reference to the source.
Author Response
Dear Reviewer,
We would like to thank you very much for your valuable comments and suggestions which were very useful and helpful to improve the manuscript quality. We revised the manuscript extensively according to the comments and suggestions. Changes were made in red to be easily followed.

Reviewer 4 Report
Comments and Suggestions for Authors
In the reviewed paper, the authors studied the chemical composition Allium iranicum seed. Since the Allium iranicum seed is being examined for the first time the manuscript provides a valuable analysis of Allium iranicum and has research value. However, the study should be expanded and perfected for scientific research, for example antimicrobial properties, etc. Section 2.3 Statistical analysis is not what statistical analysis is supposed to show. Section 3. Results and Discussion is more like Introduction almost discussion does not exist. The methods used are usual for this type of examination and has to be more clearly stated. Name of plant have to be written italic throughout the text and references. Compound names must be spelled correctly. However, the examination must be improved for the paper to be good enough to be published in this journal.
Comments on the Quality of English Language English is fine.Author Response
Dear Reviewer,
We would like to thank you very much for your valuable comments and suggestions which were very useful and helpful to improve the manuscript quality. We revised the manuscript extensively according to the comments and suggestions. Changes were made in red to be easily followed.
Comment 1: In the reviewed paper, the authors studied the chemical composition Allium iranicum seed. Since the Allium iranicum seed is being examined for the first time the manuscript provides a valuable analysis of Allium iranicum and has research value. However, the study should be expanded and perfected for scientific research, for example antimicrobial properties, etc.
Response 1: Given that this oilseed is being introduced for the first time, there is not much information about it. However, available scientific sources were used to the extent possible to develop the introduction of the article, and scientific material was added to the introduction section.
Comment 2: Section 2.3 Statistical analysis is not what statistical analysis is supposed to show.
Response 2: Correction was made in the relevant section.
Comment 3: Section 3. Results and Discussion is more like Introduction almost discussion does not exist.
Response 3: Discussion part were revised and expanded as advised.
Comment 4: The methods used are usual for this type of examination and has to be more clearly stated.
Response 4:The method section were revised and clarified as advised.
Comment 5: Name of plant have to be written italic throughout the text and references.
Response 5: The scientific name of the plant was written in italics.
Comment 6: Compound names must be spelled correctly. However, the examination must be improved for the paper to be good enough to be published in this journal.
Response 6: Compound names were corrected and the above-mentioned items were also done as recommended.

Round 2
Reviewer 1 Report
Comments and Suggestions for Authors
The manuscript has been thoroughly revised and appropriately corrected. I recommend its publication in its current form.
Author Response
Comment 1: The manuscript has been thoroughly revised and appropriately corrected. I recommend its publication in its current form.
Response 1: Authors thank you very much for your positive and encouraging comments and acceptance of this manuscript.
Reviewer 2 Report
Comments and Suggestions for Authors
The manuscript has been improved.
Author Response
Comment 1: The manuscript has been improved.
Response 1: Authors thank you very much for your positive and encouraging comments and acceptance of this manuscript.
Reviewer 3 Report
Comments and Suggestions for Authors
The article has been revised based on the comments. I offer to accept.
Author Response
The article has been revised based on the comments. I offer to accept.
Response 1: Authors thank you very much for your positive and encouraging comments and acceptance of this manuscript.
Reviewer 4 Report
Comments and Suggestions for Authors
The names of compounds must be uniform throughout the manuscript: like δ- and γ- tocopherols, not Alpha-tocopherol.
Author Response
Comment 1: The names of compounds must be uniform throughout the manuscript: like δ- and γ- tocopherols, not Alpha-tocopherol.
Response 1: Authors thank you very much for your helps in the reviewing this manuscript and valuable comments. Tocopherols name was corrected and now is uniform in whole manuscript.